# The applicability of basic preventive measures of the pandemic COVID-19 and associated factors among residents in Guraghe Zone

**Samuel Dessu**[1]*, **Tadesse Tsehay**[2]*, **Tadele Girum**[1], **Abebe Timerga**[3], **Mamo Solomon**[2], **Baye Tsegaye**[2], **Mulugeta Geremew**[4], **Biru Migora**[4], **Yibeltal Mesfin**[5], **Abdurezak Kemal**[1], **Fisha Alebel**[2], **Omega Tolosa**[2], **Shegaw Tesfa**[2], **Fedila Yasin**[1]

1 Department of Public Health, College of Medicine and Health Sciences, Wolkite University, Wolkite, Ethiopia, 2 Department of Nursing, College of Medicine and Health Sciences, Wolkite University, Wolkite, Ethiopia, 3 Department of Biomedical Sciences, College of Medicine and Health Sciences, Wolkite University, Wolkite, Ethiopia, 4 Department of Statistics, College of Natural and Computational Sciences, Wolkite University, Wolkite, Ethiopia, 5 Department of Midwifery, College of Medicine and Health Sciences, Wolkite University, Wolkite, Ethiopia

* dessusamuel@yahoo.com (SD); tadesse200912@gmail.com (TT)

**Data Availability Statement:** All relevant data are within the manuscript and its S1 Data.

## Abstract

### Introduction

Internationally, countries have reacted to the COVID-19 outbreak by introducing key public health non-pharmaceutical interventions to protect vulnerable population groups. In response to COVID-19, the Government of Ethiopia has been taking a series of policy actions beyond public health initiatives alone. Therefore, this study was aimed to assess the applicability of basic preventive measures of the pandemic COVID-19 and associated factors among the residents of Guraghe Zone from 18th to 29th September, 2020.

### Methods

Community based cross sectional study was conducted at Guraghe Zone from 18th to 29th September, 2020. Systematic random sampling method was applied among the predetermined 634 samples. Variables which had p-value less than 0.25 in bivariate analysis were considered as candidate for multivariable logistic regression model. P-value <0.05 was used as a cutoff point to determine statistical significance in multiple logistic regressions for the final model.

### Result

In this study, 17.7% (95% CI: 14.7, 20.5) of the respondents apply the basic preventive measures towards the prevention of the pandemic COVID-19. In addition, being rural resident (AOR: 4.78,; 95%CI: 2.50, 8.90), being studied grade 1–8 (AOR: 3.70; 95%CI: 1.70, 7.90), being a farmer (AOR: 4.10; 95%CI: 1.25, 13.35), currently not married (AOR: 2.20, 95%CI: 1.24, 4.06), having family size 1-3(AOR: 6.50; 95%CI: 3.21, 3.35), have no diagnosed medical illness (AOR: 6.40; 95%CI: 3.85, 10.83) and having poor knowledge (AOR: 3.50; 95%CI: 1.60, 7.40) were factors which are statistically significant in multivariable logistic regression model.

**Funding:** The authors received no specific funding for this work.

**Competing interests:** The authors have declared that no competing interests exist.

**Abbreviations:** AOR, Adjusted odd ratio; CDC, Center for Disease Control; CI, Confidence interval; COR, Crude odd ratio; COVID-19, Coronavirus Disease 2019; FMOH, Federal Ministry of Health; SARS-CoV-2, Severe acute respiratory syndrome coronavirus 2; SNNPR, South Nation Nationalities and Peoples Regional state; SPSS, Statistical package for social science; RNA, Ribonucleic acid; WHO, World Health Organization.

## Conclusion

Despite the application of preventive measures and vaccine delivery, the applicability of the pandemic COVID-19 preventive measures was too low, which indicate that the Zone is at risk for the infection. Rural residents, those who have lower educational level, farmers, non-marrieds, those who have lower family size, those who have diagnosed medical illnesses and those who have poor knowledge were prone to the infection with the pandemic COVID-19 due to the lower practice of applying the basic preventive measures. In addition, awareness creation should be in practice at all levels of the community especially lower educational classes and rural residents.

## Introduction

The pandemic of coronavirus disease 2019 (COVID-19) started in December 2019 in Wuhan, China [1]. Currently, the virus causing the disease is approaches each continent throughout the world and continues to spread at an alarming rate [2].

Globally, countries have acted up on the COVID-19 outbreak through introducing key public health non-pharmaceutical interventions to protect vulnerable population groups [3]. Lock down, washing hands frequently using soap in mashing for more than 20 seconds, maintain physical distancing, stay informed and follow advice given by healthcare professionals and seeking medical advice if develop cough or fever or experience difficulty of breathing and call in advance the center assigned for COVID-19 response were the recommended measures to reduce the transmission of COVID-19 by world health Organization (WHO) [4].

COVID-19 can results in a severe national problems. In addition; it results in severe psychological, economic crisis across the globe [5–8]. As the disease progressed, concerns regarding health, economy, and livelihood increased day-to-day. The findings of the pandemic's impact on these issues could help inform health officials and the public to provide mental health interventions to those who are in need [8].

The first case of COVID-19 in Ethiopia was detected on 13th March 2020 involving two tourists from Japan [1]. In accordance with the applied case definition and testing strategies, till 8th of April, 2021, the number of peoples infected with the pandemic COVID-19 in the world was 132,730,691, of them, 3,148, 358 were in Africa and of them, 221,544 were in Ethiopia. Among them, globally 2,880,726 were died, of them, 114,610 were in Africa and of them, 3,058 were in Ethiopia [4].

In response to COVID-19, the Government of Ethiopia has been taking a series of policy actions beyond public health initiatives alone [9]. These include closing schools, restricting use of public transportation, banning large meetings, and suspending sporting and religious gatherings. A state of emergency has been put in effect and staying at home and working from there has been strongly advised [10]. On March 2021, Ethiopia has received 2.184 million doses of the Astra Zeneca COVID-19 vaccine via the COVAX facility and the country is on immunizing the frontline workers and the most susceptible individuals [11].

Despite the implementation of such strategies, the occurrence of the pandemic COVID-19 is still increasing. Therefore, this study focus on the evaluation of the applicability of basic prevention methods of the pandemic COVID-19 may help to the populations at risk to reduce the risk of mortality.

## Methods and materials

### Study design, area and period

A community based cross-sectional study was conducted at Guraghe Zone from 18th to 29th September, 2020. Guraghe Zone is one of the 13 Zones encountered at South Nations Nationalities and Peoples regional state (SNNPR). Wolkite town is its administrative sit and located at 158 km south from Addis Ababa. Based on the 2007 Ethiopian national population and housing census, the population of the Zone is projected to be about 1,609,908 and 51.38% are females. Administratively the Zone is divided in to 16 districts (Woredas) and 4 city administrations.

### Populations

**Populations.** In this study, all the residents at the Guraghe Zone were considered as a source population and all the selected populations in the Guraghe Zone from 18th to 29th September, 2020 from the source population were termed as study populations.

### Inclusion and exclusion criteria

All the ambulatory residents of the Guraghe Zone were included into the study and residents of Guraghe Zone who cannot able to communicate (unable to listen and talk) and children who have age less than 15 were excluded from the study.

### Sample size determination and sampling procedures

The required sample size was determined using formula for single population proportion formula $n = \frac{z_{\frac{a}{2}}^2 P(1-P)}{d^2}$, Where n denotes the sample size, $z_{\frac{a}{2}}$ denotes the reliability coefficient of standard error, at 5% level of significance (which is 1.96), P = proportion of residents applicability of basic preventive measures at Guraghe zone (Which is 50% because there was no study conducted before) and d = margin of error. Therefore, the calculated sample size was 384. Using design effect for the sample in using interclass correlation (δ) = 1.5, it became 576. Through considering 10% for non-response rate, the final sample size for the study was 634.

Among the 20 districts found in Guraghe Zone, 7 districts (Abeshge Woreda, Emdebir town administration, Enemor Woreda, Edja woreda, Gumer Woreda, Meskan woreda and Wolkite town administration) were randomly selected using lottery method. From town administrations, four sub cities (two sub cities from Emdebir town administration and two sub cities from Wolkite town administration) were randomly selected using lottery method. Again from the randomly selected rural districts, 23 kebelles were randomly selected (Three kebelles from Abeshge Woreda, five Kebelles from Enemor Woreda, Four Kebelles from Edja Woreda, Five Kebelles from Gumer woreda and six kebelles from Meskan woreda).

Finally, the calculated sample size was proportionally allocated to the randomly selected districts (Abeshge (28), Emdeber (31), Enemor (156), Edja (121), Gumer (114), Meskan (137) and Wolkite town administration (47)) and the households from each district were selected using systematic random sampling method. Within each of the selected household, one individual was taken, primarily the house hold head (father) or the mother or kids based on the hierarchy (eldest to youngest).

### Study variables

The dependent variable for this study was the applicability of basic COVID-19 preventive measures and the independent variables were socio demographic variables (age, sex, residence,

educational status, occupation, marital status, religion and estimated monthly income), knowledge status of the respondents, attitude towards COVID-19, availability of Mass media and diagnosed medical health problem (Comorbidity).

## Operational definitions

**High knowledge.** If the respondent answers at least 11 of the 14 knowledge assessment questions correctly [12].

**Moderate knowledge.** If the respondent answers at least nine of the 14 knowledge assessment questions correctly [12].

**Poor knowledge.** If the respondent responds $\leq 8$ knowledge assessment questions correctly [12].

**Good knowledge.** If the respondent responds $\geq 9$ knowledge assessment questions correctly [12].

**Proper hand washing practice.** If the respondent wash his/her hands using soap and water in mashing his hands for more than 20 seconds or using hand sanitizers [13].

**Favorable attitude towards COVID-19.** a mean attitude score $\geq 30$ [12].

**Unfavorable attitude towards COVID-19.** a mean attitude score less than 30 [12].

Apply basic preventive measures: if the respondent applies complete lockdown (stay at home) or reasonable exit in wearing mask appropriately, proper hand washing practice and keeps physical distance [13].

## Data collection tool and procedures

Data collection tool was adapted from WHO resources and similar studies. Initially, it was prepared in English and was translated to Amharic by language experts to ensure consistency. Data were collected using structured interviewer administered questionnaires. Data were collected by 15-experienced BSC nurses and were supervised by three Msc holders in health science through the entire data collection process.

The knowledge questions have 14 items. These items include the participant knowledge about clinical presentations (items 1–4), transmission routes (items 5–8) and prevention and control (items 9–14) of COVID-19. Participants were given "true," "false," or "I don't know" response options to these items. A correct response to an item were assigns 1 point, while an incorrect/don't know response assigned 0 points. The maximum total score ranged from 0–14, with a higher score indicating better knowledge about COVID-19.

To measure attitudes towards COVID-19, 10 items (with minimum score 10 and maximum score 50) were used. Response of each item was recorded on 5-point Likert scale as follows strongly disagree (1-point), disagree (2-point), neutral (3-point), agree (4-point), and strongly agree (5-point). The overall level of attitude was categorized using original Bloom's cut-off point, as positive if the score was 80–100%(40–50), neutral if the score was 60–79% (30–40) and negative if the score was less than 60 (10–29). A mean score $\geq 30$ was carried out as a favorable attitude and a score less than 30 indicated an unfavorable attitude toward COVID-19.

## Data quality management

Data quality was assured by caring out careful design of data collection tool and appropriate modification was made, appropriate recruitment and one day training was given on the objective of the study, selection of study participants, how to keep confidentiality of the collected data, how to fill the data collection format and data quality management and follow-up for data collectors and supervisors. Intensive supervision was done by investigators and supervisors during the whole period of data collection.

A random sample of questionnaires were reviewed by the supervisors and the investigators to conform reliability of data before data collection and the investigators also made random cross checked for their completeness, accuracy, and consistency at the end of each day and corrective discussion was undertaken with all the research team members. Data were checked for completeness and consistency and then it was coded, entered and stored into the computer using Epi data version 3.01 statistical software. Pretest was done at Butajira town on 5% of study subjects and modification will be made accordingly. Both the validity and the reliability test were checked.

## Data processing and analysis

Data were entered into Epi-data version 3.1 and exported to SPSS version 25 for Windows, then cleaned, edited, coded and exploratory data analysis was carried out to check the levels of missing values, presence of influential outliers, multi-co linearity. Crude odd ratios and AOR were computed to assess the presence of degree of association between the outcome variable and the explanatory variables.

Both bivariate and multivariable logistic regression model were fitted to assess the association between outcome and explanatory variables. Those independent variables that had p-value less than 0.25 in bivariate analysis were entered in to the multivariable logistic regressions model. Backward stepwise regression was used for choosing determinant variables. Extent of strength was presented using odds ratios and its 95% confidence intervals. P-value <0.05 was used as a cutoff point to determine statistical significance in multiple logistic regressions for the final model. Hosmer and Lemeshow as well as omnibus test were used to test the model fitness. Multicollineaity was checked using standard error. Finally, the result was presented using texts and tables.

## Ethical consideration

Ethical clearance was obtained from Wolkite University College of medicine and health sciences ethical review board and permission letter was obtained from the Guraghe Zone health department and the corresponding Woreda health care administrative offices. A written consent was obtained directly from the respondents. All the necessary measures were taken to maintain and assure the confidentiality and all benefits of the patients.

## Results

### The socio-demographic characteristics of the respondents

The minimum and maximum age of the respondents was 18 years old and 75 years old respectively, with the mean of 33.±11.6 years. Nearly, half (49.4%) of the respondents had age between 36 to 64 years old. In this study, the number of males and females were almost equal, 50.5% and 49.5% were males and females respectively. Regarding the marital status, more than half (57.1%) of the respondents were married and maximum of the respondents (42.9%) were studied college and above.

Among the urban residents (142), only 18(12.7%) were applied the basic preventive measures. In addition, these basic preventive measures were applied by one fourth (19.1%) of the rural residents. Among respondents who studied college and above (272), these preventive measures were applied only by 41 (15.1%) and around three fourths (74.4%) of those respondents who studied grade 1–8 did not apply these preventive measures.

Regarding the marital status of the respondent, 241(38.0%), 362(57.1%), 13(2.1%) and 18 (2.8%) of the respondents were single, married, divorced and widowed respectively. Around

one fourths (25.2%) of the respondents were students and nearly half (49.4%) of the respondents have 4–6 family members within the household. More than on fifth (21.9%) of the students and around one fourth (24.3%) of the currently not employed study subjects were applied the basic preventive measures. Regarding mass media, 526 (83.0%) of the respondents have mass media. Among those who have mass media (526), 445 (84.6%) of the respondents did not apply the basic preventive measures (Table 1).

## The knowledge status of the respondents

In this study, 139(21.9%; 95%CI: 18.9, 25.1) of the respondents have good knowledge. Around 95% (606) of the respondents were aware of the main clinical symptoms of COVID-19. Regarding the knowledge towards high risk and prognosis, only 246(38.8%) of the respondents know that those who were elder, those with chronic illness and obese patients were commonly prone to sever cases. Similarly, 541(85.3%) of the respondents were aware about its transmission through respiratory droplets of infected individuals.

In addition, 594(93.7%) and 601(94.8%) of the respondents were know about wearing face mask and not touching eyes, nose and mouse as a preventive measures of the pandemic COVID-19. The basic preventive measures were applied on 12(8.6%) and 100(20.2%) of the respondents who have poor and good knowledge respectively. All of the respondents have information towards the pandemic COVID-19 preventive measures. Among them, 526 (83.0%), 83 (13.1%) and 25(3.9%) of them used mass media, health care workers and neighbors as basic source respectively (Table 2).

**Table 1. The socio-demographic characteristics of residents in Guraghe Zone, 2020 (n = 634).**

| Variables | Category | Frequency | Percent |
|---|---|---|---|
| Age (in years) | 18–35 | 205 | 32.3 |
| | 36–64 | 313 | 49.4 |
| | >64 | 116 | 18.3 |
| Residence | Urban | 142 | 22.4 |
| | Rural | 492 | 77.6 |
| Educational status | Unable to read and write | 51 | 8.0 |
| | Only read and write | 59 | 9.3 |
| | Grade 1–8 | 121 | 19.1 |
| | Grade 9–12 | 131 | 20.7 |
| | College and above | 272 | 42.9 |
| Occupational status | Farmer | 83 | 13.1 |
| | Student | 160 | 25.2 |
| | Currently not employed | 37 | 5.8 |
| | Government employed | 248 | 39.1 |
| | Private business employed | 106 | 16.7 |
| Religious status | Orthodox | 378 | 59.6 |
| | Muslim | 162 | 25.6 |
| | Protestant | 60 | 9.5 |
| | Catholic | 26 | 4.1 |
| | Others | 8 | 1.3 |
| Family size | 1–3 | 205 | 32.3 |
| | 4–6 | 313 | 49.4 |
| | >6 | 116 | 18.3 |

Table 2. Knowledge status of the respondents on COVID-19 among residents in Guraghe Zone, 2020 (n = 634).

| Knowledge assessment questions | Knowledge Category | | |
|---|---|---|---|
| | Yes | No | I don't know |
| **Knowledge of cause and symptoms** | | | |
| The main clinical symptoms of COVID-19 are fever, fatigue, dry cough, and myalgia | 606(95.6%) | 14(2.2%) | 14(2.2%) |
| Unlike the common cold, stuffy nose, runny nose, and sneezing are less common in persons infected with the COVID-19 virus | 261(41.2%) | 332(52.4%) | 41(6.5%) |
| **Knowledge of high risk and prognosis** | | | |
| Not all persons with COVID-19 will develop severe cases. Only those who are elderly, have chronic illnesses & are obese are more likely to be severe cases | 246(38.8%) | 350(55.2%) | 38(6.0%) |
| There currently is no effective cure for COVID-19, but early symptomatic and supportive treatment can help most patients recover from the infection | 188(29.7%) | 437(68.9%) | 9(1.4%) |
| **Knowledge about mode of transmission and infectiousness** | | | |
| The COVID-19 virus spreads via respiratory droplets of infected individuals | 541(85.3%) | 82(12.9%) | 11(1.7%) |
| Eating or contacting wild animals would result in the infection by the COVID-19 virus | 416(65.6%) | 154(24.3%) | 64(10.1%) |
| Persons with COVID-19 cannot infect the virus to others when a fever is not present | 112(17.7%) | 489(77.1%) | 33(5.2%) |
| **Knowledge about ways of prevention** | | | |
| Proper washing hand with soap and water is one method of preventing COVID-19 | 594(93.7%) | 40(6.3%) | 0(0%) |
| One way of prevention of COVID-19 is not touching the eye, nose by unwashed hands | 601(94.8%) | 33(5.2%) | 0(0%) |
| To prevent the infection by COVID-19, individuals should avoid going to crowded places such as train stations, market and avoid taking public transportations | 542(85.5%) | 89(14.0%) | 3(0.5%) |
| Ordinary residents can wear general medical masks to prevent the infection by the COVID- 19 virus | 593(93.5%) | 35(5.5%) | 6(0.9%) |
| People who have contact with someone infected with the COVID-19 virus should be immediately isolated in a proper place | 563(88.8%) | 69(10.9%) | 2(0.3%) |
| Isolation and treatment of people who are infected with the COVID-19 virus are effective ways to reduce the spread of the virus | 596(94.0%) | 34(5.4%) | 5.4%) |
| Children and young adults don't need to take measures to prevent the infection by the COVID-19 virus | 106(16.7%) | 501(79.0%) | 27(4.3%) |

## Attitude status of the respondents

The mean attitude score point was 39.03 points, ranging from 19 to 50 points. The majority of the participants 597(94.2%; 95%CI: 92.3, 95.9) had a favorable attitude towards the applicability of basic preventive measures of the pandemic COVID-19. More than half (52.5%) of the participants agreed positively with all attitude questions, that stated about the prevention of the pandemic COVID-19 (**Table 3**).

## Comorbid health problems

Nearly, one fifths (21.8%) of the respondents had comorbid health problems. Among them, 56 (8.8%) of them had DM, 53(8.4%) had hypertension, 16(2.5%) had HIV/AIDS and the remaining 13(2.1%) had both hypertension and DM.

## The applicability of basic preventive measures

Among those respondents, 91(14.4%) of the respondents were used lockdown as a preventive measures of the pandemic COVID-19. In addition; 21(3.3%) of the respondents did not apply lockdown but reasonable exit with the appropriate preventive measures of COVID-19. Therefore; the magnitude of respondents who apply the basic preventive measures to the pandemic COVID-19 was 17.7% (95% CI: 14.7, 20.5) (**Table 4**). The most common mentioned reason for fail to apply the basic preventive measures of the pandemic COVID-19 were poor knowledge, poor attitude and unavailability of the preventive measures (mask and hand sanitizers due to poor finance), which was 139 (78.1%), 37(5.8%) and 147 (23.2%) respectively.

**Table 3. The level of attitude towards the prevention of COVID-19 among residents in Guraghe Zone, 2020 (n = 634).**

| Attitude domain | Level of attitude | | | | |
|---|---|---|---|---|---|
| | **Strongly disagree** | **Disagree** | **Neutral** | **Agree** | **Strongly agree** |
| Do you agree that the Ethiopian government was handling the COVID-19 health crisis well? | 52(8.2%) | 126(19.9%) | 37(5.8%) | 364(57.4%) | 55(8.7%) |
| Do you agree that COVID-19 will be successfully controlled? | 37(5.8%) | 87(13.7%) | 65(10.3%) | 391(61.7%) | 54(8.5%) |
| Do you agree that Ethiopian would be able to win the battle against COVID-19? | 24(3.8%) | 38(6.0%) | 31(4.9%) | 425(67.0%) | 116(18.3%) |
| Do you agree that Self-protection is necessary for the protection of others? | 8(1.3%) | 18(2.8%) | 12(1.9%) | 402(63.4%) | 194(30.6%) |
| Do you agree that Lockdown is an effective measure to control the transmission? | 27(4.3%) | 59(9.3%) | 20(3.2%) | 334(52.7%) | 194(30.6%) |
| Do you agree that wearing face mask protects against COVID-19? | 3(0.5%) | 31(4.9%) | 5(0.8%) | 421(66.4%) | 174(27.4%) |
| Do you agree that appropriate hand washing practice is important in the prevention of COVID-19? | 5(0.8%) | 17(2.7%) | 6(0.9%) | 409(64.5%) | 197(31.1%) |
| Do you agree that the transmission of COVID-19 can be minimized in keeping physical distance? | 17(2.7%) | 37(5.8%) | 14(2.2%) | 386(60.9%) | 180(28.4%) |
| Did you agree that COVID-19 leads to stigma? | 45(7.1%) | 75(11.8%) | 21(3.3%) | 369(58.2%) | 124(19.6%) |
| Do you agree that if getting COVID-19, you will accept isolation in health facilities? | 30(4.7%) | 67(10.6%) | 14(2.2%) | 344(54.3%) | 179(28.2%) |

## Factors associated with the applicability basic preventive measures

This study assesses factors associated with the applicability of basic preventive measures of the pandemic COVID-19. In Bivariate analysis, residence, educational level, occupational status, marital status, family size, diagnosed medical health problem and knowledge status were the candidate variables for multivariable logistic regression model. In multivariable logistic regression model, residence, educational level, occupational status, marital status, family size, diagnosed medical health problem and knowledge towards the preventive measures were statistically significant factors associated with the applicability of the pandemic COVID-19 ([Table 5]).

The odd of the likelihood of not applying the basic preventive measures of the pandemic COVID-19 among rural residents were 4 times higher as compared with the urban residents (AOR: 4.78; 95%CI: 0.25, 0.89). Respondents who studied grade 1–8 were 3 times more likely not to apply the basic preventive measures as compared with those who studied college and above (AOR: 3.70; 95%CI: 1.70, 7.90). The odd of the likelihood of not applying the basic preventive measures among farmers was 4 times higher as compared with the private business employees (AOR: 4.10; 95%CI: 1.25, 13.35). Individuals currently not married were 2 times more likely not to apply the basic preventive measures as compared with currently married (AOR: 2.20; 95%CI: 1.24, 4.06).

**Table 4. The applicability of the basic preventive measures of the pandemic COVID-19 among residents in Guraghe Zone, 2020 (n = 634).**

| Variables | Applicability of preventive measures | |
|---|---|---|
| | **Yes** | **No** |
| Lockdown | 91(14.4%) | 543(85.6%) |
| Appropriate frequent hand washing | 253(39.9%) | 381(60.1%) |
| Stop hand shaking | 152(24.0%) | 482(76.0%) |
| Avoid touching eye, nose, and mouth | 110(17.4%) | 524(82.6%) |
| Using face mask as a preventive measure | 88(13.9%) | 546(86.1%) |
| Apply two meters physical distance | 63(9.9%) | 571(90.1%) |
| Avoided going to the crowed place | 79(12.5%) | 555(87.5%) |
| Using elbow to cover during coughing/sneezing | 454(71.6%) | 180(28.4%) |

**Table 5. Bivariable analysis of factors associated with the applicability of the basic preventive measures of the pandemic COVID-19 among residents in Guraghe Zone, 2020 (n = 634).**

| Variables | Category | Status | | COR (95%CI) |
|---|---|---|---|---|
| | | Not yet applied basic preventive measures | Applied basic preventive measures | |
| Age (in years) | 18–35 | 349 | 68 | 2.2(0.55, 8.72) |
| | 36–64 | 166 | 41 | 1.7(0.43, 7.00) |
| | >64 | 7 | 3 | 1 |
| Sex | Male | 267 | 53 | 1 |
| | Female | 255 | 59 | 0.85(0.57, 1.29) |
| Residence | Urban | 124 | 18 | 1 |
| | Rural | 398 | 94 | 0.62(0.36, 1.06)* |
| Educational level | Unable to read and write | 47 | 4 | 2.08(0.71, 6.10)* |
| | Only read and write | 52 | 7 | 1.32(0.56, 3.10)* |
| | Grade 1–8 | 90 | 31 | 0.51(0.30, 0.87)* |
| | Grade 9–12 | 102 | 29 | 0.62(0.37, 1.06)* |
| | College and above | 231 | 41 | 1 |
| Occupational status | Farmer | 78 | 5 | 3.62(1.29, 10.13)* |
| | Student | 125 | 35 | 0.83(0.45, 1.54) |
| | Currently not employed | 28 | 9 | 0.72(0.29, 1.77) |
| | Government employed | 205 | 43 | 1.2(0.62, 1.99) |
| | Private business employed | 86 | 20 | 1 |
| Marital status | Currently married | 284 | 78 | 1 |
| | Currently not married | 238 | 34 | 1.9(1.24, 2.98)* |
| Family size | 1–3 | 188 | 17 | 6.0(3.23, 11.30)* |
| | 4–6 | 259 | 54 | 2.62(1.62, 4.24)* |
| | >6 | 75 | 41 | 1 |
| Mass media availability | Yes | 445 | 81 | 1 |
| | No | 77 | 31 | 0.45(0.28, 0.73)* |
| Diagnosed medical health problem | Yes | 82 | 56 | 1 |
| | No | 440 | 56 | 5.36(3.46, 8.32)* |
| Knowledge status | Good | 127 | 12 | 1 |
| | Poor | 395 | 100 | 0.37(1.19, 0.70)* |
| Attitude | Favorable | 33 | 4 | 1 |
| | Unfavorable | 489 | 108 | 0.55(0.19, 1.58) |

Key note: *indicates variables which have p-value <0.25 in bivariable analysis

The odd of the likelihood of the applicability of the basic preventive measures among respondents who have a family size 1–3 is 6 times higher as compared with those who have more than six families (AOR: 6.50; 95%CI: 3.21, 13.35). Similarly, residents who have a family size 4–6 were 2 times more; likely to not to apply the basic preventive measures of the pandemic COVID-19. Respondents who did not have any diagnosed medical health problems were 6 times more likely not to apply the basic preventive measures of the pandemic COVID-19 (AOR: 6.40; 95%CI: 3.85, 10.83) (**Table 6**).

## Discussion

This study assesses the proportion of the respondents who apply the basic preventive measures of the pandemic COVID-19 and associated factors among the residents of Guraghe Zone. Considering the fact that Ethiopia is a multi-ethnic country with vastly different economic

**Table 6. Factors associated with the applicability of the basic preventive measures of the pandemic COVID-19 among residents in Guraghe Zone, 2020 (n = 634).**

| Variables | Category | COR (95%CI) | AOR(95%CI) | P-value |
|---|---|---|---|---|
| Residence | Urban | 1 | 1 | 0.0047 |
| | Rural | 0.62(0.36, 1.06)* | 4.78(2.5, 8.9)** | |
| Educational level | Unable to read and write | 2.08(0.71, 6.10)* | 1.03(0.28, 3.88) | 0.011 |
| | Only read and write | 1.32(0.56, 3.10)* | 0.42(0.13, 1.37) | |
| | Grade 1–8 | 0.51(0.30, 0.87)* | 3.7(1.70, 7.90)** | |
| | Grade 9–12 | 0.62(0.37, 1.06)* | 0.55(0.27, 1.14) | |
| | College and above | 1 | 1 | |
| Occupational status | Farmer | 3.62(1.29, 10.13)* | 4.1(1.25, 13.35)** | 0.027 |
| | Student | 0.83(0.45, 1.54) | 0.45(0.21, 1.01) | |
| | Currently not employed | 0.72(0.29, 1.77) | 0.57(0.0.19, 1.72) | |
| | Government employed | 1.2(0.62, 1.99) | 0.47(0.19, 1.01) | |
| | Private business employed | 1 | 1 | |
| Marital status | Currently married | 1 | 1 | 0.003 |
| | Currently not married | 1.9(1.24, 2.98)* | 2.2(1.24, 4.06)** | |
| Family size | 1–3 | 6.0(3.23, 11.30)* | 6.5(3.21, 13.35)** | 0.0001 |
| | 4–6 | 2.62(1.62, 4.24)* | 2.4(1.35, 4.25)** | |
| | >6 | 1 | 1 | |
| Diagnosed medical health problem | Yes | 1 | 1 | 0.0001 |
| | No | 5.36(3.46, 8.32)* | 6.4(3.85, 10.83)** | |
| Knowledge status | Good | 1 | 1 | 0.002 |
| | Poor | 0.37(1.19, 0.70)* | 3.5(1.60, 7.40) | |

Key note: *indicates variables which have p-value <0.25,

**indicates variables which have p-value <0.05

source, education levels, traditions, it is expected that the levels of knowledge, attitude, and applicability of basic preventive measures will also markedly differ in the population.

The proportion of residents who apply the basic preventive measures to the pandemic COVID-19 was 17.7% (95% CI: 14.7, 20.5). This study finding was less than the study conducted at Nepal (78.9%), Addis Zemen Hospital, Northwest Ethiopia (52.5%) and Northwest Gonder, Ethiopia (59.4%) [14–16]. This might be due to the difference in the socio economic status of the countries. In low income countries, the individuals did not apply the lockdown to sustain their daily lives and they cannot invest any cost in purchasing the personal protective devices [17].

The magnitude of residents who have good knowledge was 21.9%. This study finding is less than the study conducted at Nepal, Uganda and Henan China; which was 76%, 69% and 89% respectively [9, 14, 18]. This discrepancy in knowledge level may be related to the better preparedness for the worst and the number of residents who were severely affected with the infection especially in Henan China.

The proportion of individuals who have favorable attitude towards COVID-19 prevention methods was 94.2%. This study finding was greater than the study conducted at Nepal was 54.7% [14] and which was less than the study conducted at China (98%) [10]. These discrepancies might be due to the residents' knowledge level, level of health information dissemination, government implementation policies and the residents' adherence towards the prevention methods.

Consistent with the study conducted in Bangladesh [19], the odd of the likelihood of not applying the basic preventive measures of the pandemic COVID-19 among rural residents

were 4 times higher as compared with the urban residents (AOR: 4.78; 95%CI: 0.25, 0.89). This might be probably due to the inaccessibility of the social and mass media especially television across the households, where media plays an important protective role through raising the public awareness about protective measures and through the countering rumors [20].

In addition; the urban populations were more prone to the information disseminations regarding the preventive measures of the pandemic COVID-19. In addition, the rural residents are prone to the cultural prejudices as compared with the urban residents [21]. Information sources are more available and spread faster in cities, and people can obtain first-hand information quickly as compared with the rural residents [22].

Inconsistent with the study conducted at Nepal and Bangladesh [14, 19], respondents who studied grade 1–8 were 3 times more likely not to apply the basic preventive measures as compared with those who studied college and above (AOR: 3.70; 95%CI: 1.70, 7.90). This is commonly due to lower grade studies were prone to lower knowledge and a weak learning ability, making it harder to grasp the relevant knowledge regarding COVID-19, unable to adopt a protective attitude and be less positive. It has been suggested that health education should be targeted at people with different educational levels and different needs for health education. For the less educated population, easy-to-understand publicity materials may be more effective.

Occupational status of the respondents became one of the factors which affect the applicability of the basic preventive measures of the pandemic COVID-19. The odd of the likelihood of fail applying the basic preventive measures among farmers was 4 times higher as compared with the private business employees (AOR: 4.10; 95%CI: 1.25, 13.35). Farmers did not adopt recommended measures to promote safe and consistent use of preventive measures, lock down to sustain their life and did not use face mask (which became uncomfortable during farming) [23].

Individuals currently not married were 2 times more likely not to apply the basic preventive measures as compared with currently married (AOR: 2.20; 95%CI: 1.24, 4.06). This might be due to the reason that, single individuals are less likely to have family support and who did not be able to stay at home to win their daily lives as compared with the married one.

The odd of the likelihood of the applicability of the basic preventive measures among respondents who have a family size 1–3 is 6 times higher as compared with those who have more than six families (AOR: 6.50; 95%CI: 3.21, 13.35) and residents who have a family size 4–6 were 2 times more; likely to fail to apply the basic preventive measures of the pandemic COVID-19 as compared with those who have more than six families (AOR: 2.40; 95%CI: 1.35, 4.25). This might be due to that having increased number of family members have better encouragement and awareness acquired through different sources. Information missed through one member can be acquired through the other member, which can be an input for the applicability of the preventive measures of the pandemic COVID-19.

Respondents who did not have any diagnosed medical health problems were 6 times more likely fail to apply the basic preventive measures of the pandemic COVID-19 (AOR: 6.40; 95% CI: 3.85, 10.83). Patients who have an experience of infection of chronic diseases have more compliance with preventive behaviors and are more careful in the prevention of diseases [20]. In addition, it might be due to that, individuals who have chronic diseases have information about its feature of being serious illness on those who have a chronic diseases and elder populations.

Knowledge became one of the associated factors for the applicability of the basic prevention measures of the pandemic COVID-19. Individuals who have poor knowledge were 3 times more likely to fail to apply the basic preventive measures of the pandemic COVID-19 (AOR: 3.50; 95% CI: 1.60, 7.40). This is due to knowledgeable individuals are more careful and responsible for the application of the preventive measures. In addition, knowledgeable individuals know the way of prevention, mode of transmission and the risk if it was not prevented appropriately.

## Conclusion

Despite the application of preventive measures and vaccine delivery, the applicability of the pandemic COVID-19 preventive measures was too low, which indicate that the Zone is at risk for the infection. Rural residents, those who have lower educational level, farmers, non-marrieds, those who have lower family size, those who have diagnosed medical illnesses and those who have poor knowledge were prone to the infection with the pandemic COVID-19 due to the lower practice of applying the basic preventive measures. In addition, awareness creation should be in practice at all levels of the community especially lower educational classes and rural residents.

## Recommendations

- Health professionals should act on awareness creation regarding the practice for all the community especially lower educational classes.

- Health facilities and health offices should follow and encourage individuals who have lower family size and a follow up link should be created between them and health extension practitioners and health facilities.

- Health extension practitioners should give special emphasis and encourage rural residents to increase their practice level

- The government should supply the basic preventive measures such as mask and sanitizers for financially poor individuals

- Residents must practice the information gained through different sources.

- Residents who have no diagnosed medical illnesses should apply the basic preventive measures.

## Supporting information

**S1 Data.**
(SAV)

## Acknowledgments

We would like to thank the mothers who are directly involved in the study and administrator of each hospital for their effort and permission to conduct the study.

## Author Contributions

**Conceptualization:** Samuel Dessu, Tadesse Tsehay.

**Data curation:** Samuel Dessu, Tadesse Tsehay.

**Formal analysis:** Samuel Dessu, Tadesse Tsehay.

**Funding acquisition:** Samuel Dessu, Tadesse Tsehay.

**Investigation:** Samuel Dessu, Tadesse Tsehay.

**Methodology:** Samuel Dessu, Tadesse Tsehay.

**Project administration:** Tadele Girum, Abebe Timerga, Mamo Solomon, Baye Tsegaye, Abdurezak Kemal.

**Resources:** Tadele Girum, Abebe Timerga, Mamo Solomon, Baye Tsegaye, Abdurezak Kemal.

**Software:** Tadele Girum, Abebe Timerga, Mamo Solomon, Baye Tsegaye, Abdurezak Kemal.

**Supervision:** Tadele Girum, Abebe Timerga, Mamo Solomon, Baye Tsegaye, Abdurezak Kemal.

**Validation:** Tadele Girum, Abebe Timerga, Mamo Solomon, Baye Tsegaye, Abdurezak Kemal.

**Visualization:** Mulugeta Geremew, Biru Migora, Yibeltal Mesfin, Fisha Alebel, Omega Tolosa.

**Writing – original draft:** Mulugeta Geremew, Biru Migora, Yibeltal Mesfin, Fisha Alebel, Omega Tolosa.

**Writing – review & editing:** Shegaw Tesfa, Fedila Yasin.

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
