## [Decision Letter · Decision Letter 0]

31 Mar 2021

PONE-D-21-05518

The applicability of basic preventive measures of the pandemic COVID-19 and associated factors among residents in Guraghe Zone

PLOS ONE

Dear Dr. Dessu,

Thank you for submitting your manuscript to PLOS ONE. After careful consideration, we feel that it has merit but does not fully meet PLOS ONE’s publication criteria as it currently stands. Therefore, we invite you to submit a revised version of the manuscript that addresses the points raised during the review process.

You must provide a reason why the effect size 1.5 is used. To due this the sample size is much lower and so the conclusion drawn from the analysis is questionable. 

We look forward to receiving your revised manuscript.

Kind regards,

Enamul Kabir

Academic Editor

PLOS ONE

Journal Requirements:

Please provide additional details regarding participant consent. In the ethics statement in the Methods and online submission information, please ensure that you have specified (1) whether consent was suitably informed and (2) what type you obtained (for instance, written or verbal). If your study included minors under age 18, state whether you obtained consent from parents or guardians. If the need for consent was waived by the ethics committee, please include this information.

Reviewers' comments:

Reviewer's Responses to Questions

**Comments to the Author**

1. Is the manuscript technically sound, and do the data support the conclusions?

Reviewer #1: Yes

Reviewer #2: Yes

Reviewer #3: No

2. Has the statistical analysis been performed appropriately and rigorously? 

Reviewer #1: No

Reviewer #2: Yes

Reviewer #3: No

3. Have the authors made all data underlying the findings in their manuscript fully available?

Reviewer #1: Yes

Reviewer #2: Yes

Reviewer #3: Yes

4. Is the manuscript presented in an intelligible fashion and written in standard English?

Reviewer #1: No

Reviewer #2: Yes

Reviewer #3: Yes

5. Review Comments to the Author

Reviewer #1: 1. The manuscript has a lot of typo errors grammatical, spelling, punctuation, and consistency in word usage like; COVID-19, COVID-2019 and COVID 19, Guraghe zone and Gurage zone, dyed instead of died.

2. High knowledge: If the respondent answers 11 of the 14 knowledge assessment questions correctly (10). What if the respondent answer 12, 13, and 14 of 14 knowledge assessment questions correctly????

3. Moderate knowledge: If the respondent answers nine of the 14 knowledge assessment questions correctly (10). What if the respondent answers 10 of 14 knowledge assessment questions correctly????

4. Poor or low knowledge: If the respondent responds below nine knowledge assessment questions correctly (10).

5. Good knowledge: If the respondent responds above nine knowledge assessment questions correctly

6. How did you measure attitude is not operationalized

7. The maximum total score ranged from 0–13, with a higher score indicating better knowledge about COVID-19. How can it be the maximum score of 13 if one individual answers all 14 questions correctly the maximum score will be 14? So, how do you justify it?

8. Data were cleaned, edited, coded, and entered into Epi-data version 3.1 and exported to SPSS version 25 for Windows. How can you cleaned, edited, and coded before data entry?

9. Television is not social media?

10. How do you manage the multicollinearity between being a rural residence and a lower educational level?

11. Should supply the basic preventive measures such as mask and sanitizers for financially poor individuals: You didn’t assess the availability of these preventive accessories so how can you recommend.

12. Those independent variables that had p-value less than 0.25 in bivariate analysis were entered in to the multivariable logistic regressions model. What is your justification to use 0.25 as cut off points?

Reviewer #2: In this manuscript, the authors conducted a community based cross-sectional study to assess The applicability of basic preventive measures of the pandemic COVID-19 and associated factors among residents in Guraghe Zone. The study is well written, is easy to follow and covers a hot topic, but some issues should be improved before publication.

Comments

1.The study is well thought off. I believed that the topic and the content of the manuscript was different. So, it will be advisable to modify the title like KAP

2.In the method section, replace Method by Method and material

3.There are many language mistakes, please revisit the manuscript for correction.

4. Reference for your operational definition?

5.Please give some explanations about the current availability of the vaccine

6.Please complete all necessary information on the title of each Table

7. Discussion section: Will be useful to the reader to add some interesting recent literature about the updates against COVID-19 outbreak and related tools to counteract the same

8.Used very few reference, which results poor interpretation of your result, please use the following reference (Akalu Y, Ayelign B, Molla MD. Knowledge, attitude and practice towards COVID-19 among chronic disease patients at Addis Zemen Hospital, Northwest Ethiopia. Infection and drug resistance. 2020;13:1949., Mulu GB, Mittiku YM, Jemere BA. Preparedness and Approaches of Healthcare Providers to Tackle the Transmission of Covid-19 among North Shewa Zone Hospitals, Amhara, Ethiopia, 2020.,Shibabaw T, Teferi B. Knowledge and Practice Toward Prevention of SARS-CoV-2 Among Healthcare Workers at Delghi Primary Hospital During a Massive Test Campaign in Northwest Gondar, Ethiopia: Institution-Based Descriptive Cross-Sectional Survey. Infection and Drug Resistance. 2021;14:381. )

9.Conclusion Section: The paragraph requires a general revision to eliminate redundant sentences and please refine and don’t repeat it in the abstract part.

Reviewer #3: major

3 why use design effect 1.5? , is there scientifically recommended to use design effect 1.5

minor

1, there is sentence and paragraph without reference on introduction part

2, study are description part, please use recent data not more than 5 years

3, the sampling procedure is multistage , it is better represent graphically that makes easily understand to readers

4. on table 5 there is missing data , please incorporate this data

5, it is better to exclude those data are not significant on multi-variate regression

6, please include the p-value for those factors that are significant at multi-variant

6. PLOS authors have the option to publish the peer review history of their article (what does this mean?). If published, this will include your full peer review and any attached files.

Reviewer #1: No

Reviewer #2: **Yes: **Birhanu Ayelign

Reviewer #3: No

---

## [Author Response · Author response to Decision Letter 0]

16 Apr 2021

Response to the Reviewers

1. Is the manuscript technically sound, and do the data support the conclusions?

Reviewer #1: Yes

Reviewer #2: Yes

Reviewer #3: No

Response; Thank you and certain revision were made.

2. Has the statistical analysis been performed appropriately and rigorously?

Reviewer #1: No

Reviewer #2: Yes

Reviewer #3: No

Response: Thank you and it was revised in detail.

3. Have the authors made all data underlying the findings in their manuscript fully available?

Reviewer #1: Yes

Reviewer #2: Yes

Reviewer #3: Yes

Response: Thank you

4. Is the manuscript presented in an intelligible fashion and written in standard English?

Reviewer #1: No

Reviewer #2: Yes

Reviewer #3: Yes

Response: Thank you and it was revised by a language expert. 

5. Review Comments to the Author

Response: It was revised and we have responded point by point for each raised concerns and corrected as highlighted in the revised version.

Reviewer #1: 

1. The manuscript has a lot of typo errors grammatical, spelling, punctuation, and consistency in word usage like; COVID-19, COVID-2019 and COVID 19, Guraghe zone and Gurage zone, dyed instead of died.

• Response: Thank you. All the inconsistencies were resolved and the whole manuscript was revised by language expert. 

2. High knowledge: If the respondent answers 11 of the 14 knowledge assessment questions correctly (10). What if the respondent answer 12, 13, and 14 of 14 knowledge assessment questions correctly????

Response: Thank you and it was revised and it was to mean at least 11 of the 14 assessment questions. 

3. Moderate knowledge: If the respondent answers nine of the 14 knowledge assessment questions correctly (10). What if the respondent answers 10 of 14 knowledge assessment questions correctly????

• Response: It was revised and which was to mean at least nine of the knowledge assessment questions. 

4. Poor knowledge: If the respondent responds below nine knowledge assessment questions correctly (10).

• Response: It was revised as:- “Poor knowledge: If the respondent responds < 8 knowledge assessment questions correctly”. 

5. Good knowledge: If the respondent responds above nine knowledge assessment questions correctly

• Response: It was revised as: - “Good knowledge: If the respondent responds > 9 knowledge assessment questions correctly”.

6. How did you measure attitude is not operationalized

• Response: It was mentioned at the end of data collection tool and procedure but no we have stated at the subtitle, operational definition. 

7. The maximum total score ranged from 0–13, with a higher score indicating better knowledge about COVID-19. How can it be the maximum score of 13 if one individual answers all 14 questions correctly the maximum score will be 14? So, how do you justify it?

• Response: Thank you. It was a typing error, which was to mean 14. If a respondent answers all the knowledge assessment questions correctly, stated as scored 14 of the 14 questions. 

8. Data were cleaned, edited, coded, and entered into Epi-data version 3.1 and exported to SPSS version 25 for Windows. How can you cleaned, edited, and coded before data entry?

Response: The statement was rephrased and stated as:- “Data were entered into Epi-data version 3.1 and exported to SPSS version 25 for Windows, then cleaned, edited, coded and exploratory data analysis was carried out to check the levels of missing values, presence of influential outliers, multi-co linearity”.

9. Television is not social media?

Response: It was revised and which was to mean mass media.

10. How do you manage the multicollinearity between being a rural residence and a lower educational level?

Response: All the variables were checked for multicholinearity but multicholinearity was not existed. Therefore; no any management is required unless multicholinearity was observed. 

11. Should supply the basic preventive measures such as mask and sanitizers for financially poor individuals: You didn’t assess the availability of these preventive accessories so how can you recommend.

Response: It was missed during manuscript preparation and incorporated now. This study was conducted as a baseline for further studies and for conducting community service across the study area. This study was presented within the university and community service was delivered through delivering the sanitizers, mask and health information dissemination. 

12. Those independent variables that had p-value less than 0.25 in bivariate analysis were entered in to the multivariable logistic regressions model. What is your justification to use 0.25 as cut off points?

Response: In this study, we had used 12 independent variables. If we had sufficient variables, we can minimize the cutoff point but if the number of variables were not much we can increase the cutoff point. In addition; increasing the cutoff point will keep the marginally significant variables. Therefore; we have used the cutoff point 0.25 to select the candidate variable for multivariable analysis. 

Reviewer #2: 

In this manuscript, the authors conducted a community based cross-sectional study to assess The applicability of basic preventive measures of the pandemic COVID-19 and associated factors among residents in Guraghe Zone. The study is well written, is easy to follow and covers a hot topic, but some issues should be improved before publication.

1) The study is well thought off. I believed that the topic and the content of the manuscript was different. So, it will be advisable to modify the title like KAP

Response: It was not aimed to investigate the knowledge and attitude but both of them were independent variables. They were described and as independent factors associated with the applicability of the basic preventive measures of COVID-19. As you have seen in the introduction part, it was focused to show the gap in the applicability of the basic preventive measures. In addition, as you have seen the outcome variable is the applicability of the basic preventive measures of the pandemic COVID-19, not KAP. 

2) In the method section, replace Method by Method and material

Response: It was replaced accordingly.

3) There are many language mistakes, please revisit the manuscript for correction.

Response: The whole manuscript was revised by a language expert. 

4) Reference for your operational definition?

Response: Thank you all the operational definitions were cited. 

5) Please give some explanations about the current availability of the vaccine

Response; It was incorporated at the introduction section.

6) Please complete all necessary information on the title of each Table

Response: Thank you. It was revised and all the necessary information was incorporated. 

7) Discussion section: Will be useful to the reader to add some interesting recent literature about the updates against COVID-19 outbreak and related tools to counteract the same

Response: Certain updated recent articles were cited. 

8) Used very few reference, which results poor interpretation of your result, please use the following reference (Akalu Y, Ayelign B, Molla MD. Knowledge, attitude and practice towards COVID-19 among chronic disease patients at Addis Zemen Hospital, Northwest Ethiopia. Infection and drug resistance. 2020;13:1949., Mulu GB, Mittiku YM, Jemere BA. Preparedness and Approaches of Healthcare Providers to Tackle the Transmission of Covid-19 among North Shewa Zone Hospitals, Amhara, Ethiopia, 2020.,Shibabaw T, Teferi B. Knowledge and Practice Toward Prevention of SARS-CoV-2 Among Healthcare Workers at Delghi Primary Hospital During a Massive Test Campaign in Northwest Gondar, Ethiopia: Institution-Based Descriptive Cross-Sectional Survey. Infection and Drug Resistance. 2021;14:381. )

Response: Thank you. All of them were cited. 

9) Conclusion Section: The paragraph requires a general revision to eliminate redundant sentences and please refine and don’t repeat it in the abstract part.

Response: The conclusion was revised and certain amendments were made.

Response to Reviewer #3

Why use design effect 1.5? , is there scientifically recommended to use design effect 1.5

• Response: Design effect is determined by the researcher in considering the heterogeneity of the population. Most researchers use design effect 2 but it is possible to use also 1.5. We have decided to use it 1.5 in considering, the heterogeneity of the populations (which is adequate for them, there is no extreme heterogeneity between them) and the cost that we have afford. 

Minor

1. There is sentence and paragraph without reference on introduction part

Response: Thank you. All the statements were cited. 

2. On table 5 there is missing data , please incorporate this data

Response: It was not a missing data, which was for the variables which were not statistically significant in multivariable analysis but to resolve this, we have the table in to two (table 5 & 6). 

3. It is better to exclude those data are not significant on multi-variate regression

Response: It was revised and removed. 

4. Please include the p-value for those factors that are significant at multi-variant

Response: Thank you, it was included. 

---

## [Decision Letter · Decision Letter 1]

15 Jun 2021

PONE-D-21-05518R1

The applicability of basic preventive measures of the pandemic COVID-19 and associated factors among residents in Guraghe Zone

PLOS ONE

Dear Dr. Dessu,

Thank you for submitting your manuscript to PLOS ONE. After careful consideration, we feel that it has merit but does not fully meet PLOS ONE’s publication criteria as it currently stands. Therefore, we invite you to submit a revised version of the manuscript that addresses the points raised during the review process.

Please address second reviewer concern and rephrase sentences in the abstract and conclusion to minimize repetition. 

We look forward to receiving your revised manuscript.

Kind regards,

Enamul Kabir

Academic Editor

PLOS ONE

Journal Requirements:

Reviewers' comments:

Reviewer's Responses to Questions

**Comments to the Author**

1. If the authors have adequately addressed your comments raised in a previous round of review and you feel that this manuscript is now acceptable for publication, you may indicate that here to bypass the “Comments to the Author” section, enter your conflict of interest statement in the “Confidential to Editor” section, and submit your "Accept" recommendation.

Reviewer #1: All comments have been addressed

Reviewer #2: All comments have been addressed

Reviewer #3: All comments have been addressed

2. Is the manuscript technically sound, and do the data support the conclusions?

Reviewer #1: Yes

Reviewer #2: Yes

Reviewer #3: Yes

3. Has the statistical analysis been performed appropriately and rigorously? 

Reviewer #1: Yes

Reviewer #2: Yes

Reviewer #3: Yes

4. Have the authors made all data underlying the findings in their manuscript fully available?

Reviewer #1: Yes

Reviewer #2: Yes

Reviewer #3: Yes

5. Is the manuscript presented in an intelligible fashion and written in standard English?

Reviewer #1: Yes

Reviewer #2: Yes

Reviewer #3: Yes

6. Review Comments to the Author

Reviewer #1: Line space in abstract part is not consistent. So make it consistent the line space for the conclusion part is not consistent with the other parts`

Reviewer #2: The author addresses all comments, however, still they did not understood one of my comment regarding redundancy of sentence the conclusion in both the abstract and main body. Therefore, please rephrase and rewrite the conclusion to minimize repetition

Reviewer #3: (No Response)

7. PLOS authors have the option to publish the peer review history of their article (what does this mean?). If published, this will include your full peer review and any attached files.

Reviewer #1: No

Reviewer #2: No

Reviewer #3: No

---

## [Author Response · Author response to Decision Letter 1]

15 Jun 2021

Both concdrns were addressed and corrected.

---

## [Decision Letter · Decision Letter 2]

21 Jul 2021

PONE-D-21-05518R2

The applicability of basic preventive measures of the pandemic COVID-19 and associated factors among residents in Guraghe Zone

PLOS ONE

Dear Dr. Dessu,

Thank you for submitting your manuscript to PLOS ONE. After careful consideration, we feel that it has merit but does not fully meet PLOS ONE’s publication criteria as it currently stands. Therefore, we invite you to submit a revised version of the manuscript that addresses the points raised during the review process.

One of the reviewers raised some minor issues those need to be fixed before taking final decision.

We look forward to receiving your revised manuscript.

Kind regards,

Enamul Kabir

Academic Editor

PLOS ONE

Journal Requirements:

Additional Editor Comments (if provided):

Reviewers' comments:

Reviewer's Responses to Questions

**Comments to the Author**

1. If the authors have adequately addressed your comments raised in a previous round of review and you feel that this manuscript is now acceptable for publication, you may indicate that here to bypass the “Comments to the Author” section, enter your conflict of interest statement in the “Confidential to Editor” section, and submit your "Accept" recommendation.

Reviewer #1: All comments have been addressed

Reviewer #2: All comments have been addressed

Reviewer #3: All comments have been addressed

2. Is the manuscript technically sound, and do the data support the conclusions?

Reviewer #1: Yes

Reviewer #2: Yes

Reviewer #3: Yes

3. Has the statistical analysis been performed appropriately and rigorously? 

Reviewer #1: Yes

Reviewer #2: Yes

Reviewer #3: Yes

4. Have the authors made all data underlying the findings in their manuscript fully available?

Reviewer #1: Yes

Reviewer #2: Yes

Reviewer #3: Yes

5. Is the manuscript presented in an intelligible fashion and written in standard English?

Reviewer #1: Yes

Reviewer #2: Yes

Reviewer #3: Yes

6. Review Comments to the Author

Reviewer #1: Comments

The authors have carefully addressed almost all the issues raised by reviewers in the first review process. However, the additional comments below should be addressed to enhance the quality of the paper.

On Abstract, data processing and analysis, and result session better to address these comments which are highlighted in the main manuscript:

• P-value ≤0.05 better to change <0.05

• All prevalence, proportion, and magnitude are better to be in one decimal point including its confidence intervals. For instance, 21.9% (95%CI: 18.9, 25.1) of the respondents have good knowledge, 94.2% (95%CI: 92.3, 95.9) had a favorable attitude, and 17.7% (95% CI: 14.7, 20.5) apply basic preventive measures …………

• The odds ratio and its respective confidence interval better to be in two decimal points. For example, (AOR: 4.78; 95%CI: 2.50, 8.90)

• As a principle; the prevalence, proportion, magnitude, odds, and its respective confidence interval should have a similar decimal points which means one decimal point for prevalence, proportion, and magnitude and two decimal points for odds.

• All the recommendations have no owner it doesn’t tell anything about for whom you are going to recommend. So, it is better to indicate the specific stakeholders for each of your recommendations.

Reviewer #2: The author addressed all comments provided by me and all other reviewers. Therefore, I confirmed that it is accepted.

Reviewer #3: the author addressed all comments adequately and proper statically analysis and technically good and I recommend for publication

7. PLOS authors have the option to publish the peer review history of their article (what does this mean?). If published, this will include your full peer review and any attached files.

Reviewer #1: No

Reviewer #2: No

Reviewer #3: No

---

## [Author Response · Author response to Decision Letter 2]

22 Jul 2021

Response to reviewers

• P-value ≤0.05 better to change <0.05 

Response: It was revised and corrected as “P-value <0.05 was used as a cutoff point to determine statistical significance in multiple logistic regressions for the final model”. 

• All prevalence, proportion, and magnitude are better to be in one decimal point including its confidence intervals. For instance, 21.9% (95%CI: 18.9, 25.1) of the respondents have good knowledge, 94.2% (95%CI: 92.3, 95.9) had a favorable attitude, and 17.7% (95% CI: 14.7, 20.5) apply basic preventive measures …………

Response: It was revised and corrected all over the manuscript as:- “In this study, 17.7% (95% CI: 14.7, 20.5) of the respondents apply the basic preventive measures towards the prevention of the pandemic COVID-19”. 

• The odds ratio and its respective confidence interval better to be in two decimal points. For example, (AOR: 4.78; 95%CI: 2.50, 8.90)

o Response: It was revised and corrected as:- “In addition, being rural resident (AOR: 4.78,; 95%CI: 2.50, 8.90), being studied grade 1-8 (AOR: 3.70; 95%CI: 1.70, 7.90), being a farmer (AOR: 4.10; 95%CI: 1.25, 13.35), currently not married (AOR: 2.20, 95%CI: 1.24, 4.06), having family size 1-3(AOR: 6.50; 95%CI: 3.21, 3.35), have no diagnosed medical illness (AOR: 6.40; 95%CI: 3.85, 10.83) and having poor knowledge (AOR: 3.50; 95%CI: 1.60, 7.40) were factors which are statistically significant in multivariable logistic regression model”.

• As a principle; the prevalence, proportion, magnitude, odds, and its respective confidence interval should have a similar decimal points which means one decimal point for prevalence, proportion, and magnitude and two decimal points for odds. 

o Response: It was revised and corrected as per your recommendations.

• All the recommendations have no owner it doesn’t tell anything about for whom you are going to recommend. So, it is better to indicate the specific stakeholders for each of your recommendations.

o Response: It was revised and the recommendations were given for the specific bodies.

---

## [Decision Letter · Decision Letter 3]

11 Aug 2021

The applicability of basic preventive measures of the pandemic COVID-19 and associated factors among residents in Guraghe Zone

PONE-D-21-05518R3

Dear Dr. Dessu,

We’re pleased to inform you that your manuscript has been judged scientifically suitable for publication and will be formally accepted for publication once it meets all outstanding technical requirements.

Kind regards,

Enamul Kabir

Academic Editor

PLOS ONE

Additional Editor Comments (optional):

Reviewers' comments:

Reviewer's Responses to Questions

**Comments to the Author**

1. If the authors have adequately addressed your comments raised in a previous round of review and you feel that this manuscript is now acceptable for publication, you may indicate that here to bypass the “Comments to the Author” section, enter your conflict of interest statement in the “Confidential to Editor” section, and submit your "Accept" recommendation.

Reviewer #1: All comments have been addressed

Reviewer #2: All comments have been addressed

2. Is the manuscript technically sound, and do the data support the conclusions?

Reviewer #1: Yes

Reviewer #2: Yes

3. Has the statistical analysis been performed appropriately and rigorously? 

Reviewer #1: Yes

Reviewer #2: Yes

4. Have the authors made all data underlying the findings in their manuscript fully available?

Reviewer #1: Yes

Reviewer #2: Yes

5. Is the manuscript presented in an intelligible fashion and written in standard English?

Reviewer #1: Yes

Reviewer #2: Yes

6. Review Comments to the Author

Reviewer #1: All comments have been addressed; The language, typographic errors, analysis, and overall the paper addressed scientific writeup formats. And all my concerns are addressed and it is fit to be accepted for possible publication.

Reviewer #2: All comments are well addressed and incorporated in the main body of the manuscript. So, please do not submit again more for revision

7. PLOS authors have the option to publish the peer review history of their article (what does this mean?). If published, this will include your full peer review and any attached files.

Reviewer #1: No

Reviewer #2: No

---

## [Editor Report · Acceptance letter]

16 Aug 2021

PONE-D-21-05518R3 

The applicability of basic preventive measures of the pandemic COVID-19 and associated factors among residents in Guraghe Zone 

Dear Dr. Dessu:

I'm pleased to inform you that your manuscript has been deemed suitable for publication in PLOS ONE. Congratulations! Your manuscript is now with our production department. 

Kind regards, 

on behalf of

Dr. Enamul Kabir 

Academic Editor

PLOS ONE